# HeliCon: Dual-Level Contrastive Alignment for Robust Medical VQA under Long-Tailed Distribution

## Abstract

Learning robust multimodal representations for medical visual question answering (Med-VQA) is challenging due to the imbalanced distribution of semantic concepts. Frequent clinical patterns form robust embedding structures, while rare yet clinically important concepts often yield fragile representations, hindering reliable prediction. To address this issue, we propose HeliCon, a dual-level contrastive alignment framework that follows a conceptual "double helix" structure. It intertwines two complementary mechanisms: (1) memory banks at the instance and prototype levels, which preserve sample diversity while enforcing semantically meaningful clustering; and (2) contrastive learning objectives at the hard and soft levels, which refine head embeddings and transfer relational knowledge to tail concepts. Collectively, these mechanisms enhance the learning of robust and semantically consistent multimodal representations across both frequent and rare concepts. At inference, a retrieval-augmented mechanism further enriches contextual information integration by leveraging relevant answer embeddings from the training set. Experiments on Med-VQA benchmarks demonstrate that HeliCon achieves improved performance, with a particularly 3.51% absolute gain over the state-of-the-art on the PathVQA dataset, highlighting its effectiveness in producing robust representations under long-tailed data distributions. The code is available at: https://github.com/DeepModeler/medical-vqa-helicon.git

## 1 Introduction

Medical Visual Question Answering (Med-VQA) requires constructing robust multimodal representations that capture the intricate interactions between medical images and textual queries. Despite notable progress in modeling such interactions, Med-VQA remains particularly challenging due to the limited availability of labeled data and the highly imbalanced distribution of clinical concepts (Nguyen et al., 2019; Khanal et al., 2024). Many prior works train VQA models with token-level autoregressive supervision (Zhan et al., 2020; Van Sonsbeek et al., 2023; Do et al., 2021), which provides limited semantic guidance for structuring the multimodal embedding space. This limitation is especially severe in short-answer scenarios, where a small number of tokens cannot adequately encode the semantic context necessary for meaningful similarity comparisons. To address this, contrastive pre-training has been explored for vision–language alignment (Li et al., 2023b; Liu et al., 2021a; Yan et al., 2024), yielding improved embedding robustness and semantic consistency. However, most of these approaches rely on instance-level objectives that favor frequent (head) samples while neglecting rare (tail) concepts, limiting understanding about rare but clinically important concepts.

Beyond data scarcity and long-tailed distributions, another crucial challenge in VQA is data bias. Extensive studies show that VQA models often exploit spurious correlations between questions and answers, relying on language priors instead of genuine multimodal reasoning (Cho et al., 2023; Liu et al., 2024; Niu et al., 2021; Zhan et al., 2023; Xu et al., 2025; Si et al., 2022). To mitigate this limitation, debiasing strategies have been proposed from multiple perspectives, including generative adversarial modeling of biases (Cho et al., 2023), causal intervention to eliminate fine-grained language priors (Liu et al., 2024), causal graph modeling to reduce cross-modal confounders (Xu et al., 2025), counterfactual reasoning (Niu et al., 2021; Zhan et al., 2023), and contrastive learning with biased samples (Si et al., 2022). Although these approaches improve performance under

distribution shifts by mitigating question-driven biases, they remain limited in enhancing the reliability of rare concept embeddings and providing structural guidance in the multimodal representation space. Moreover, the related problem of answer distribution bias remains largely underexplored, where frequent answers dominate the embedding space while rare yet clinically crucial concepts fail to form robust representations, leading to poor generalization.

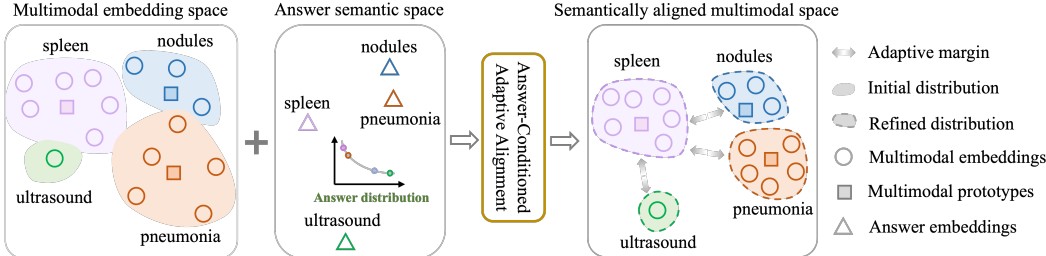

Figure 1: Illustration of adaptive embedding alignment, where tail sample embeddings are refined via answer-conditioned alignment and head sample embeddings are optimized through dual-level alignment, resulting in a more structured and semantically coherent embedding space.

Our work complements these studies by shifting the focus from question bias to answer bias. The key insight of our approach is that the relationships between head and tail answer embeddings can facilitate representation learning, with information from frequent concepts guiding the alignment of rare concepts, as illustrated in Figure 1. We propose a dual-level contrastive alignment framework that structures the multimodal embedding space. Frequent concept embeddings are optimized using both instance-level and prototype-level objectives, yielding consistent and well-structured representations. Rare concept embeddings are softly aligned toward head prototypes via answer-conditioned guidance, which transfers relational knowledge to enhance their robustness. This process is supported by two complementary memory banks, including a sample-level bank that preserves instance diversity and a prototype-level bank that enhances semantic clustering structure. At inference, we further refine answer generation with a retrieval-augmented mechanism that retrieves semantically relevant answer embeddings from the training set and integrates them with test sample embeddings, enriching contextual representations for more accurate answer generation. This framework ensures robustness to long-tailed distributions in medical VQA dataset.

## 2 RELATED WORKS

### 2.1 BIAS AND LONG-TAIL CHALLENGES IN VQA TASK

VQA models are prone to over-reliance on superficial question-answer cues, favoring language priors over visual grounding (Ramakrishnan et al., 2018; Cadene et al., 2019; Ruggeri et al., 2023; Si et al., 2022; Jung et al., 2024; Seth et al., 2023). To mitigate this, prior works have explored adversarial regularization to penalize language priors (Ramakrishnan et al., 2018), feature-level debiasing via representation adjustment (Ruggeri et al., 2023; Jung et al., 2024), reweighting or modifying biased samples (Cadene et al., 2019; Si et al., 2022), and residual-based feature refinement (Seth et al., 2023). Beyond this general bias, the long-tailed answer distribution presents an additional challenge, with frequent head concepts dominating training while rare tail concepts remain underrepresented. Methods to address this issue include adaptive data calibration (Song et al., 2025), retrieval-augmented learning for rare concepts (Parashar et al., 2024), long-tailed aware mixture-of-experts routing (Cai et al., 2025), and hierarchical modeling for generalization (Portelli et al., 2022). While these methods focus on mitigating question-driven biases, the medical domain poses additional challenges due to scarce annotated data and long-tailed answer distributions (He et al., 2020). Existing long-tail research in medical domain mainly focuses on classification tasks (Zheng et al., 2024), with limited exploration in medical VQA tasks. Our work shifts attention to answer-driven bias, leveraging a dual-level contrastive learning framework to improve robustness of rare sample embeddings and enforce a semantically structured multimodal embedding space.

### 2.2 CONTRASTIVE REPRESENTATION LEARNING IN MEDICAL VQA TASK

Contrastive learning has emerged as a powerful paradigm for multimodal representation learning. CLIP (Radford et al., 2021) has shown strong transferability across diverse visual tasks through large-

scale image–text pretraining, paving the way for multimodal representation learning. In the medical domain, contrastive learning methods have been adapted to address data scarcity and domain-specific challenges. For instance, PMC-CLIP (Lin et al., 2023) and BiomedCoOp (Koleilat et al., 2025) extend CLIP-style training to the medical domain by leveraging large-scale biomedical corpora or prompt learning for efficient adaptation. In addition, outcome-aware contrastive learning has been proposed to enhance survival analysis with improved calibration (Lee et al., 2024). These medical adaptations highlight the effectiveness of contrastive supervision in learning robust and transferable representations under limited annotation. Recently, contrastive learning strategies have been extended to medical VQA task. PubMedCLIP (Eslami et al., 2023) fine-tunes CLIP on medical domain data. CPCR (Liu et al., 2022) uses contrastive pretraining to enhance feature extractor. MUMC (Li et al., 2023b) leverages both unimodal and multimodal contrastive objectives to enhance image–text alignment. MENDER (Lin et al., 2025) combine contrastive learning with generative diffusion to handle difference-aware scenarios. These studies highlight contrastive learning as a key mechanism enabling medical VQA models to be robust and generalizable under complex reasoning demands.

## 3 METHODOLOGY

Our framework structures the multimodal embedding space by explicitly aligning input question-image embeddings with answer embeddings using a memory-augmented contrastive scheme. As illustrated in Figure 2, the model maintains two memory banks: a sample-level memory, which stores instance embeddings and is updated by replacing entries with batch embeddings, and a prototype-level memory, which holds class prototypes and is updated via momentum-based moving average of batch prototypes. Training samples are divided into head and tail sets based on answer frequency. For head samples, we employ dual-level contrastive learning objectives: an instance-level loss preserves the diversity of individual samples, while a prototype-level loss encourages embeddings to cluster around their corresponding prototypes. Tail samples are softly aligned to the head prototypes through a prototype-level soft contrastive learning objective, which transfers relational knowledge from head to tail samples based on answer similarities. During training, contrastive learning objectives are computed between batch samples and memory banks, with gradients only flowing through the batch embeddings. Finally, these objectives are integrated with the standard autoregressive generation loss to jointly learn an embedding space that is both robust and semantically structured.

### 3.1 MEMORY-AUGMENTED MULTIMODAL EMBEDDINGS

Let $\mathcal{E}(\cdot)$ and $\mathcal{G}(\cdot)$ denote the text token embedder and the decoder of LLM, and $\mathcal{V}(\cdot)$ and $\mathcal{P}(\cdot)$ denote the visual encoder and projector, respectively. For the $i$-th training sample with question $Q_i$, image $I_i$, and answer $A_i$, we construct the multimodal input and answer embeddings as:

$$\mathcal{X}_i = [\mathcal{E}(Q_i); \mathcal{P}(\mathcal{V}(I_i))] \in \mathbb{R}^{(l_Q+l_I)\times d}, \quad \mathcal{Y}_i = \mathcal{E}(A_i) \in \mathbb{R}^{l_A \times d}, \tag{1}$$

where $[\cdot; \cdot]$ denotes concatenation of embeddings. We denote the set of training sample embeddings as $\mathcal{D} = \{(\mathcal{X}_i, \mathcal{Y}_i)\}_{i=1}^N$, and the set of answer categories as $\mathcal{C}$. For notational clarity, we reuse $\mathcal{D}$ to indicate corresponding representations derived from these samples (e.g., after LLM decoding or aggregation), depending on context. Then, the multimodal input and answer embeddings are combined and fed into the LLM to obtain the last hidden states:

$$H_i = \mathcal{G}([\mathcal{X}_i; \mathcal{Y}_i]) \in \mathbb{R}^{(l_Q+l_I+l_A)\times d}, \tag{2}$$

which are then split into multimodal embeddings and answer embeddings:

$$\mathcal{X}_i' = H_i[1 : l_Q + l_I, :] \in \mathbb{R}^{(l_Q+l_I)\times d}, \quad \mathcal{Y}_i' = H_i[l_Q + l_I + 1 : l_Q + l_I + l_A, :] \in \mathbb{R}^{l_A \times d}. \tag{3}$$

We aggregate token-level embeddings using either mean pooling or a query-based attention network, where the latter employs a learnable query vector to attend over the sequence via multi-head attention, yielding compact embeddings:

$$x_i = \phi(\mathcal{X}_i') \in \mathbb{R}^d, \quad y_i = \phi(\mathcal{Y}_i') \in \mathbb{R}^d, \tag{4}$$

where $\phi(\cdot)$ denotes the aggregation function. To address the long-tailed answer distribution, we partition the training set into head and tail subsets based on answer frequency. Let $\mathcal{H}$ denote head set, whose sample answers occur more than a threshold $n_{\text{thresh}}$, and $\mathcal{T}$ tail set, whose sample answers occur less than $n_{\text{thresh}}$:

$$\mathcal{H} = \{(x_i, y_i) \mid n_{c_i} \geq n_{\text{thresh}}\}, \quad \mathcal{T} = \{(x_i, y_i) \mid n_{c_i} < n_{\text{thresh}}\}, \tag{5}$$

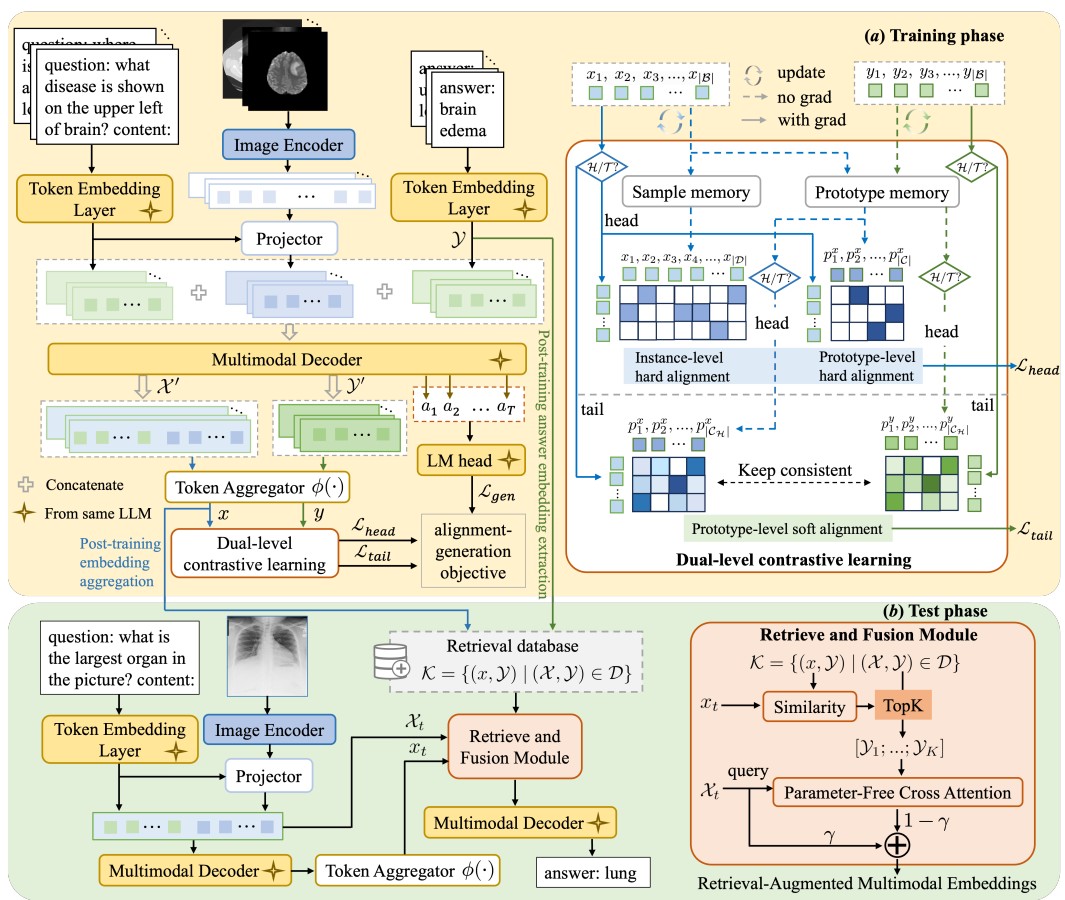

Figure 2: Overview of the proposed dual-level contrastive alignment framework. (a) Training phase. (b) Test phase.

where $n_c$ is the number of occurrences of answer category $c$ in $\mathcal{D}$. Then, we construct two memory banks. *Sample-level memory* $\mathcal{M}_s$ stores instance-level input embeddings $x$ and is updated dynamically by replacing entries with the current batch $\mathcal{B}$:

$$\mathcal{M}_s \leftarrow \text{update}(\mathcal{M}_s, \{x_i \mid x_i \in \mathcal{B}\}). \tag{6}$$

*Prototype-level memory* $\mathcal{M}_p$ stores category-level prototypes for each answer class $c$, consisting of two components: the input aggregated multimodal prototype $p_c^x$ and the corresponding answer prototype $p_c^y$. Both are initialized as the mean of the embeddings of all training samples belonging to each answer class, $p_c^x = \frac{1}{|\mathcal{D}_c|} \sum_{x \in \mathcal{D}_c} x, p_c^y = \frac{1}{|\mathcal{D}_c|} \sum_{y \in \mathcal{D}_c} y$, and updated with a momentum-based moving average:

$$p_c^x \leftarrow \alpha p_c^x + (1-\alpha)\bar{x}_c, \quad p_c^y \leftarrow \alpha p_c^y + (1-\alpha)\bar{y}_c, \tag{7}$$

where $\alpha$ is a momentum factor, $\bar{x}_c = \frac{1}{|\mathcal{B}_c|} \sum_{x \in \mathcal{B}_c} x$ and $\bar{y}_c = \frac{1}{|\mathcal{B}_c|} \sum_{y \in \mathcal{B}_c} y$ denote the mean input embedding and mean answer embedding of class $c$ in the current batch, respectively.

### 3.1.1 DUAL-LEVEL HARD CONTRASTIVE LEARNING FOR HEAD SAMPLES

Head samples are trained with dual objectives: an instance-level hard contrastive loss and a prototype-level hard contrastive loss. For a head sample $x_i \in \mathcal{B} \cap \mathcal{H}$, the instance-level objective is formulated as a multi-positive InfoNCE loss computed over the union of the current batch and the memory bank:

$$\ell_i^{inst} = -\log \frac{\sum_{x_p \in \mathcal{B} \cup \mathcal{M}_s, y_a = y_i} \exp(x_i \cdot x_p / \tau)}{\sum_{x_a \in \mathcal{B} \cup \mathcal{M}_s} \exp(x_i \cdot x_a / \tau)}, \tag{8}$$

and the prototype-level objective is formulated as a single-positive InfoNCE loss (Oord et al., 2018):

$$\ell_i^{proto} = -\log \frac{\exp(x_i \cdot p_{c_i}^x / \tau)}{\sum_{c \in \mathcal{C}_{\mathcal{H}}} \exp(x_i \cdot p_c^x / \tau)}, \tag{9}$$

where $\mathcal{C}_{\mathcal{H}}$ denotes the set of categories corresponding to head answers, and $\tau$ is a temperature hyperparameter. We combine the instance-level and prototype-level losses with a weighting factor $\beta$:

$$\mathcal{L}_{head} = \sum\nolimits_{x_i \in \mathcal{B} \cap \mathcal{H}} \beta \ell_i^{inst} + (1 - \beta) \ell_i^{proto}. \tag{10}$$

This dual-level scheme leverages complementary strengths of both objectives: the instance-level loss preserves local diversity among samples, while the prototype-level loss encourages embeddings to cluster around their prototypes, jointly enhancing global semantic structure and knowledge transfer to tail samples.

### 3.1.2 PROTOTYPE-LEVEL SOFT CONTRASTIVE LEARNING FOR TAIL SAMPLES

For a tail sample $x_j \in \mathcal{B} \cap \mathcal{T}$, we perform a soft alignment with head prototypes, guided by the similarities of their corresponding answer embeddings. Specifically, we derive the predicted distribution from $x_j$ with respect to head multimodal prototypes, and the target distribution from $y_j$ with respect to head answer prototypes:

$$\hat{s}_{jk}^x = \frac{\exp(x_j \cdot p_k^x / \tau)}{\sum_{c \in \mathcal{C}_{\mathcal{H}}} \exp(x_j \cdot p_c^x / \tau)}, \quad s_{jk}^y = \frac{\exp(y_j \cdot p_k^y / \tau)}{\sum_{c \in \mathcal{C}_{\mathcal{H}}} \exp(y_j \cdot p_c^y / \tau)}, \quad k \in \mathcal{C}_{\mathcal{H}}. \tag{11}$$

Then, the prototype-level soft contrastive loss is formulated as a Kullback–Leibler divergence (Hinton et al., 2015) between the predicted distribution and the target distribution:

$$\mathcal{L}_{tail} = \sum\nolimits_{x_j \in \mathcal{B} \cap \mathcal{T}} KL(\hat{s}_{jk}^x \,\|\, s_{jk}^y). \tag{12}$$

This loss facilitates the transfer of relational knowledge from head prototypes to tail samples, yielding robust representations.

### 3.1.3 ALIGNMENT-GENERATION OBJECTIVE

The final training objective is a weighted combination of the autoregressive generation loss $\mathcal{L}_{gen}$ (Vaswani et al., 2017), head contrastive loss and tail contrastive loss:

$$\mathcal{L}_{AGO} = \mathcal{L}_{gen} + \lambda_1 \mathcal{L}_{head} + \lambda_2 \mathcal{L}_{tail}, \quad \mathcal{L}_{gen} = -\sum_{i=1}^{|\mathcal{B}|} \sum_{t=1}^{T} \log \mathcal{G}(a_{it} \mid \mathcal{X}_i, a_{i,<t}), \tag{13}$$

where $\lambda_1$ and $\lambda_2$ control the relative contributions of the dual-level contrastive objectives. This training objective encourages head samples to preserve diversity, tail samples to benefit from knowledge transfer, and the embedding space to remain semantically consistent and robust.

### 3.2 RETRIEVAL-AUGMENTED INFERENCE

After training, we construct a retrieval database $\mathcal{K} = \{(x_i, \mathcal{Y}_i) \mid i = 1, ..., |\mathcal{D}|\}$ from the entire training set, storing aggregated input embeddings $x_i$ and their corresponding answer token embeddings $\mathcal{Y}_i$. During inference, given a test sample $\mathcal{X}_t$, we first compute its aggregated multimodal embedding $x_t$, then we retrieve the top-$K$ most similar entries from the retrieval database $\mathcal{K}$ based on the similarity between $x_t$ and the stored aggregated embeddings. The selected answer token embeddings are concatenated to form the retrieved embeddings $\mathcal{Y}^{ret}$:

$$\mathcal{R}_t = \text{TopK}\left(\{(x_i, \mathcal{Y}_i) \in \mathcal{K} \mid \text{sim}(x_t, x_i)\}\right), \quad \mathcal{Y}^{ret} = [\mathcal{Y}_k]_{(x_k, \mathcal{Y}_k) \in \mathcal{R}_t}, \tag{14}$$

where $\text{sim}(\cdot, \cdot)$ denotes cosine similarity. Then the test embedding $\mathcal{X}_t$ is augmented with the retrieved embeddings via a parameter-free cross-attention module $\mathcal{CA}$, and combined with the original embedding using a weighting factor $\gamma$:

$$\tilde{\mathcal{X}}_t = \gamma \mathcal{X}_t + (1 - \gamma)\mathcal{CA}(\mathcal{X}_t, \mathcal{Y}^{ret}). \tag{15}$$

Finally, the retrieval-augmented embedding $\tilde{\mathcal{X}}_t$ is fed into the decoder for autoregressive answer generation. This procedure integrates knowledge from training set, enhancing contextual representation and facilitating accurate generation, particularly for rare concept samples.

## 4 EXPERIMENTS

### 4.1 DATASETS

For the Med-VQA task, we evaluate HeliCon on three widely used benchmarks: VQA-RAD (Lau et al., 2018), SLAKE (Liu et al., 2021b), and PathVQA (He et al., 2020), containing 3,064, 14,028, and 32,799 question–answer pairs, respectively. These datasets cover both open-ended and close-ended questions, providing diverse evaluation scenarios for training and evaluating our method.

### 4.2 IMPLEMENTATION DETAILS

We use CLIP-ViT/B-16 (Radford et al., 2021) as the visual encoder, four large language models, namely GPT2 (1.5B) (Radford et al., 2019), StableLM (1.6B) (Bellagente et al., 2024), Mistral (7B) (Albert Q. Jiang, 2023), and Llama2 (7B) (Touvron et al., 2023), as decoders, and a two-layer MLP with GELU activation as the projection module. Prior to fine-tuning on Med-VQA benchmarks, we conduct a pre-training stage using 600K image–text pairs filtered from the PMC-15M dataset released by LLaVA-Med (Li et al., 2023a), which comprises a diverse collection of biomedical image–caption pairs. The pre-training objective is image–text matching, which aligns visual and textual representations and equips the model with domain-specific multimodal knowledge. This initialization is then leveraged for fine-tuning on Med-VQA datasets. During medical VQA task training, the temperature parameter $\tau$ is set to 0.5, and the weighting factors $\lambda_1$ and $\lambda_2$ are both set to 0.5. Training is conducted for 30 epochs with a batch size of 32, using the AdamW optimizer with a learning rate of $2 \times 10^{-5}$. To ensure stable convergence in the early phase, we apply 500 warmup steps. All experiments are carried out on two NVIDIA RTX A6000 GPUs, each equipped with 48 GB of VRAM.

### 4.3 COMPARISON WITH THE STATE-OF-THE-ART

Table 1: Performance comparison with previous state-of-the-art methods on VQA-RAD, SLAKE and PathVQA datasets. For generative methods, recall is reported for open-ended questions and accuracy for closed-ended questions. For discriminative methods, accuracy is used for both question types. The best results are highlighted in bold, and the second-best results are underlined.

| Method | VQA-RAD | | SLAKE | | PathVQA | |
|---|---|---|---|---|---|---|
| | Open | Closed | Open | Closed | Open | Closed |
| *Discriminative methods* | | | | | | |
| MEVF-BAN (Zhan et al., 2020) | 49.20 | 77.20 | 77.80 | 79.80 | – | – |
| PubMedCLIP (Eslami et al., 2023) | 60.10 | 80.00 | 78.40 | 82.50 | – | – |
| M3AE (Chen et al., 2022) | 67.23 | 83.46 | 80.31 | 87.82 | – | – |
| CP+BAN+CR (Liu et al., 2022) | 60.50 | 80.40 | 80.50 | 84.10 | – | – |
| MMQL (Chen et al., 2024) | 64.25 | 85.66 | 84.19 | 90.63 | – | – |
| PMC-CLIP (Lin et al., 2023) | 67.00 | 84.00 | 81.90 | 88.00 | – | – |
| BiomedCLIP (Zhang et al., 2023) | 67.60 | 79.80 | 82.50 | 89.70 | – | – |
| MMQ (Do et al., 2021) | 53.70 | 75.80 | – | – | 13.40 | 84.00 |
| MUMC (Li et al., 2023b) | 71.50 | 84.20 | – | – | 39.00 | 90.40 |
| *Generative methods* | | | | | | |
| Prefix T. Medical LM (GPT2) (Van Sonsbeek et al., 2023) | – | – | 84.30 | 82.10 | 40.00 | 87.00 |
| FAVP (Llama2) (Yu et al., 2025) | 68.10 | **89.00** | 85.60 | 87.90 | – | – |
| LLaVA-Med (BioMed CLIP) (Li et al., 2023a) | 64.75 | 83.09 | 87.11 | 86.78 | 39.60 | 91.09 |
| Med-MoE (Phi2) (Jiang et al., 2024) | 58.55 | 82.72 | 85.06 | 85.58 | 34.74 | **91.98** |
| Med-MoE (StableLM) (Jiang et al., 2024) | 50.08 | 80.07 | 83.16 | 83.41 | 33.79 | 91.30 |
| HeliCon (GPT2) | 62.78 | 84.27 | 87.30 | 87.90 | 39.39 | 88.64 |
| HeliCon (StableLM) | 63.43 | 83.83 | 87.13 | **88.25** | 36.41 | 91.80 |
| HeliCon (Mistral) | 67.89 | 86.24 | 88.45 | 88.14 | **43.51** | 88.10 |
| HeliCon (Llama2) | **68.92** | 86.71 | **88.92** | 87.33 | 42.83 | 89.76 |

As shown in Table 1, HeliCon outperforms prior approaches on open-ended questions across all Med-VQA datasets, which are generally more challenging than closed-ended questions. Notably,

compared with MUMC (Li et al., 2023b), a method based on multiple contrastive learning objectives, HeliCon achieves gains of 2.51% and 1.40% on VQA-RAD and PathVQA closed-ended questions, respectively. PathVQA is a much larger dataset with complex open-form question–answer pairs, our method surpasses the state-of-the-art on PathVQA open-ended questions by 3.51%. These results demonstrate that HeliCon provides superior understanding ability for open-ended medical VQA tasks. Although direct comparisons with classification-based models are limited by differing evaluation metrics, HeliCon exhibits robust generation performance, producing answers that match or even exceed those from discriminative approaches without being constrained to predefined answer sets.

## 4.4 ABLATION STUDY AND ANALYSIS

**Ablation study of contrastive learning strategies.** In this section, we conduct an ablation study on the SLAKE dataset to evaluate the contributions of the key components in the HeliCon framework, including the sample-level soft contrastive learning ($SSC$) objective, the prototype-level hard contrastive learning ($PHC$) objective, and their combination in the dual-level contrastive strategy. In this experiment, the retrieval-augmented inference (RAI) module is not included. The baseline follows the standard VQA training setup in multimodal LLMs, where the question, image features, and answer are formatted in a prompt-like concatenation (i.e., `question: <question embeddings> content: <image features> answer: <answer embeddings>`) and fed to a decoder-only language model trained with a standard auto-regressive objective, resembling the supervised training scheme used in LLaVA-Med (Li et al., 2023a). Table 2 presents both quantitative and qualitative performance for each learning strategy. The results indicate that incorporating the $SSC$ objective yields the largest performance gain. This aligns with the t-SNE visualization of input multimodal embeddings, where previously entangled samples become more separated. For instance, unrelated samples corresponding to answers like "lung" and "black" or "brain" and "mri" become more separable, forming clearer answer-specific clusters. Adding the $PHC$ objective further improves performance, and the t-SNE plot show more semantically coherent clustering, with samples corresponding to similar answers (e.g., "lung," "liver," "abdomen") positioned closer together. Finally, the dual-level contrastive learning strategy achieves the best overall performance, producing an embedding space with highly structured and semantically meaningful organization. These results demonstrate that each component contributes uniquely to robust representation learning: $SSC$ enhances local sample discrimination, $PHC$ refines global prototype alignment, and their combination in the dual-level strategy benefits both head and tail concepts.

Table 2: Ablation study of learning strategies on the SLAKE dataset. $SSC$, $PHC$, and $dualC$ denote sample-level soft contrastive learning, prototype-level hard contrastive learning, and dual-level contrastive learning, respectively, while $LR$ and $CA$ denote a linear and a cross-attention projector.

| Projector | Baseline | | +SSC | | +PHC | | dualC | |
|---|---|---|---|---|---|---|---|---|
| | Accuracy | Recall | Accuracy | Recall | Accuracy | Recall | Accuracy | Recall |
| $LR$ | 86.08 | 85.71 | 86.79 | 87.25 | 86.86 | 88.03 | 87.12 | 88.56 |
| $CA$ | 85.57 | 84.33 | 86.25 | 86.69 | 86.43 | 87.46 | 86.76 | 88.01 |
| $t\text{-}SNE$ |  | |  | |  | |  | |

**Head–tail performance and embedding correlation analysis.** To assess HeliCon's ability to improve tail sample performance and semantic alignment of multimodal embeddings, we conduct quantitative evaluations on PathVQA, VQA-RAD, and SLAKE, while embedding correlation analysis is performed exclusively on PathVQA. We partition head and tail subsets using an adaptive frequency threshold $n_{thresh}$, determined such that head classes cover approximately 60% of the open-set training samples. This dynamic threshold ensures that the head/tail split naturally reflects the data distribution rather than relying on an arbitrary fixed value. The answer distributions for each dataset are provided

in Appendix Figure 7. We report performance on head, tail, and overall samples, shown in Table 3 for PathVQA and Table 4 for SLAKE and VQA-RAD. Across all datasets, HeliCon consistently outperforms the baseline, indicating its effectiveness in structuring and regularizing the embedding space, particularly for tail samples. For PathVQA and VQA-RAD, improvements on tail samples are relatively smaller than those on head samples, primarily due to the inherent difficulty of tail classes. Data statistics from these datasets indicate that tail samples span a substantially larger number of answer categories compared to head samples, and a large portion of tail answers are unseen during training. Therefore, even modest gains on tail samples are particularly significant, reflecting the effectiveness of HeliCon in handling sparse and challenging data. Notably, for SLAKE dataset, tail samples even show larger improvements than head samples.

Table 3: Performance and embedding analysis for the open-ended samples on the PathVQA dataset. It reports head and tail performance (Recall, F1-score, BLEU1) for both baseline and HeliCon methods. Additionally, it includes scatter plots of multimodal input embedding similarity versus answer embedding similarity, along with correlation metrics such as regression coefficient $\beta$, coefficient of determination $R^2$, and F-test statistic. $\mathcal{H}$, $\mathcal{T}$ and $\mathcal{O}$ denote the head, tail, and all sample sets, respectively. All F-tests for both the baseline and our method yield extremely small p-values ($p < 10^{-16}$). Bold numbers indicate larger effect sizes (higher $\beta$ or $R^2$) compared with the baseline.

| Metric | Baseline | | | HeliCon | | |
|---|---|---|---|---|---|---|
| | Head | Tail | Overall | Head | Tail | Overall |
| Recall | 53.06 | 21.05 | 39.57 | 56.47 | 23.77 | 42.69 |
| F1-score | 51.01 | 26.36 | 40.62 | 56.16 | 27.18 | 43.95 |
| BLEU1 | 59.60 | 53.74 | 57.13 | 61.07 | 55.30 | 58.64 |
| Scatter plot |  | | |  | | |

| Cases | $\beta \uparrow$ | $R^2 \uparrow$ | $F$-test | $\beta \uparrow$ | $R^2 \uparrow$ | $F$-test |
|---|---|---|---|---|---|---|
| $\mathcal{H} - \mathcal{H}$ | 0.4378 | 0.3379 | 20416 | **0.4637** | **0.5753** | 54175 |
| $\mathcal{H} - \mathcal{T}$ | 0.5656 | 0.2606 | 14096 | **0.6271** | **0.6619** | 78313 |
| $\mathcal{T} - \mathcal{T}$ | 0.6171 | 0.2639 | 7495 | **0.8744** | **0.9036** | 196088 |
| $\mathcal{O} - \mathcal{O}$ | 0.5336 | 0.2839 | 223032 | **0.5572** | **0.4715** | 501612 |

**PathVQA dataset statistics:**
Train head ratio: 60.04%, Head C/N: 170 / 5973, Tail C/N: 3053 / 3976
Test head ratio: 57.86%, Head Seen/C/N: 142 / 142 / 1950, Tail Seen/C/N: 280 / 1155 / 1420
Seen/C/N indicates the number of answer classes covered by the training data (Seen), the number of answer classes (C), and the number of samples (N).

We further analyze embedding alignment on the PathVQA dataset by computing pairwise similarities among all samples for $\mathcal{X}$ and $\mathcal{Y}$. Scatter plots shown in Table 3 demonstrate that HeliCon exhibits a strong linear correlation between $\mathcal{X}$–$\mathcal{X}$ and $\mathcal{Y}$–$\mathcal{Y}$ similarities, with points more concentrated and following a clearer linear trend, indicating well-aligned multimodal representations and a semantically consistent embedding space. While the baseline distribution is more scattered with weaker correlation. To quantify these relationships, we compute regression coefficients (Franzese et al., 2018) to measure the strength and direction of the correlations, the coefficient of determination (Chicco et al., 2021) to capture the proportion of variance in answer similarity explained by input embedding similarity, and F-test (Sanderson & Windmeijer, 2016) to assess statistical significance. Analyses are performed for "head-to-head", "head-to-tail", "tail-to-tail", and "all-to-all" samples. All F-tests yield very small p-values, indicating that the observed correlations are statistically significant. Both qualitative and quantitative analyses confirm that HeliCon effectively captures the semantic structure

of the multimodal space, particularly improving alignment for tail samples and enhancing robust generalization across long-tailed answer distribution.

Table 4: Performance comparison between Baseline and HeliCon on head, tail, and overall samples for the VQA-RAD and SLAKE datasets.

| Metric | VQA-RAD | | | | | | SLAKE | | | | | |
|---|---|---|---|---|---|---|---|---|---|---|---|---|
| | Baseline | | | HeliCon | | | Baseline | | | HeliCon | | |
| | Head | Tail | Overall | Head | Tail | Overall | Head | Tail | Overall | Head | Tail | Overall |
| Recall | 79.66 | 58.34 | 65.80 | 85.11 | 59.73 | 68.61 | 95.00 | 72.37 | 85.45 | 97.03 | 76.91 | 88.54 |
| F1-score | 77.86 | 59.94 | 66.21 | 84.70 | 61.92 | 69.89 | 94.81 | 72.54 | 85.41 | 96.79 | 76.93 | 88.41 |
| BLEU1 | 68.59 | 58.15 | 61.80 | 73.40 | 60.65 | 65.11 | 88.47 | 64.42 | 78.32 | 90.95 | 64.04 | 79.59 |

**VQA-RAD dataset statistics:**
Train head ratio: 60.03%, Head C/N: 138 / 811, Tail C/N: 330 / 540
Test head ratio: 35.00%, Head Seen/C/N: 25 / 25 / 70, Tail Seen/C/N:54 / 93 / 130

**SLAKE dataset statistics:**
Train head ratio: 60.80%, Head C/N: 18 / 1968, Tail C/N: 199 / 1269
Test head ratio: 57.79%, Head Seen/C/N: 18 / 18 / 408, Tail Seen/C/N: 109 / 111 / 298

Moreover, we conducted additional experiments with varying head ratios (60%, 70%, and 80%). The results are presented in Tables 5–7 and illustrated in Figure 8 in the Appendix 6.1.

**Effect of memory bank on contrastive learning.** To validate the effectiveness of the memory bank mechanism, we conduct experiments by varying the batch size in the contrastive learning process. When no memory bank is used, contrastive objectives are limited to within-batch comparisons, whereas the memory bank enables alignment across the entire training set. We evaluate performance with batch sizes of 8, 16, and 32. As shown in Figure 3, both accuracy and recall fall below the baseline when the batch size is 8, suggesting that contrastive learning with a small number of samples can be detrimental to the model. As the batch size increases, performance improves, particularly when the memory bank is employed for global sample contrastive learning. These results confirm that enlarging the pool of positive and negative samples via the memory bank provides richer contrastive signals, leading to more robust representation learning.

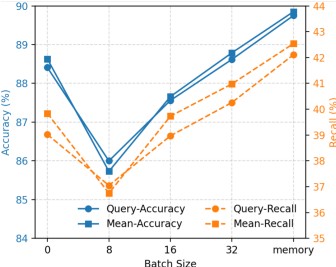

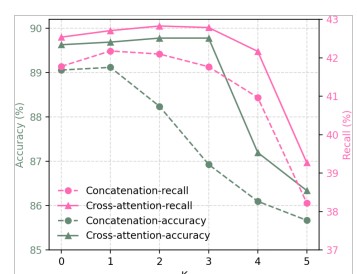

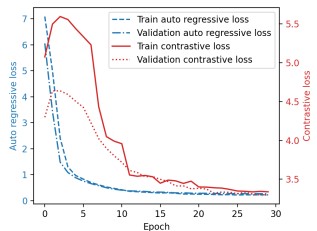

Figure 3: Performance of HeliCon under varying batch sizes for contrastive learning, where 0 corresponds to the baseline model without contrastive learning.

Figure 4: Performance of HeliCon under varying top $K$ for retrieved samples, where $K = 0$ corresponds to the setting without retrieval-augmented inference.

Figure 5: Training and validation curves of HeliCon, with all losses exhibiting stable convergence. The autoregressive and contrastive losses are shown in different colors.

**Effect of retrieval-augmented inference.** We analyze the effect of retrieval-augmented inference by varying the number of retrieved samples, $K \in \{1, 2, 3, 4, 5\}$. Two integration strategies are evaluated. In the concatenation strategy, retrieved embeddings are appended directly to the test embeddings. In the fusion strategy, retrieved answer embeddings are combined with test embeddings through a parameter-free cross-attention mechanism. As shown in Figure 4, fusion consistently outperforms concatenation, achieving its best performance at $K = 2$, while concatenation reaches its optimum at $K = 1$. Building on this analysis, we examine the performance gains contributed by the retrieval-augmented inference (RAI) module across all datasets. Figure 10 shows the improvements before and after incorporating RAI. Overall, the contribution of this module is relatively modest, with performance gains ranging from approximately 0.1% to 0.4% across the datasets.

To further illustrate the benefits of retrieval-augmented inference, we present results on four examples: two radiology images and two pathology images, each containing one head and one tail sample. In

Figure 6, we compare model outputs before and after retrieval-augmented inference and observe that retrieval enables the model to refine its predictions and produce the correct answers. Moreover, we visualize the attention of answer embeddings over image patches, demonstrating that retrieval-augmented embeddings focus more accurately on the regions relevant to the question and answer. These results indicate that incorporating correctly retrieved context helps refine predictions.

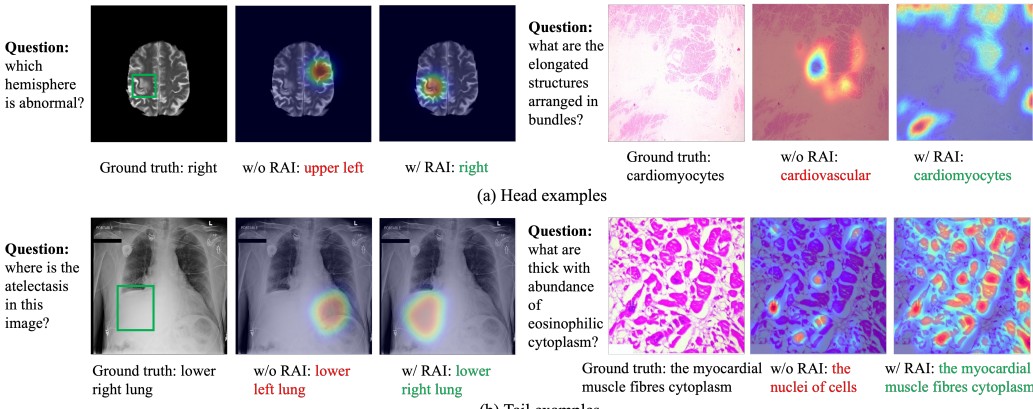

(a) Head examples

(b) Tail examples

Figure 6: Comparison of model predictions and attention maps over image regions before and after retrieval-augmented inference (RAI). Radiology images are from the SLAKE dataset, and pathology images are from the PathVQA dataset. Predictions are highlighted in green for correct answers and in red for incorrect ones.

**Hyperparameter analysis of loss weights $\lambda_1$ and $\lambda_2$** In our main experiments, we set $\lambda_1$ and $\lambda_2$ to 0.5, prioritizing the primary auto-regressive objective to ensure that sequence generation and reasoning are guided predominantly by the core task. The auxiliary losses act as supportive signals, gently shaping learning without overpowering the main objective, thereby maintaining stability and effectiveness during training. To further investigate the influence of these hyperparameters, we conducted a hyperparameter sweep on the validation set of each dataset. Figure 9 (a) shows that the head and tail losses contribute comparably to the overall loss, which motivates the use of a shared weight. For simplicity, we therefore assigned the same value to both $\lambda_1$ and $\lambda_2$. Candidate values were drawn from the set $\{0, 0.5, 1.0, 1.5, 2.0\}$. The results, presented in Figure 9 (b), indicate that across most datasets and settings, $\lambda_1 = \lambda_2 = 0.5$ achieved the best or near-best performance, supporting its use as a shared hyperparameter for all datasets.

**Convergence analysis of HeliCon.** We also analyze the optimization dynamics of HeliCon by plotting the training and validation curves of the autoregressive and contrastive losses, as shown in Figure 5. Both training and validation losses exhibit consistent trends, indicating stable convergence of the joint optimization. Notably, the autoregressive loss decreases rapidly during training, whereas the contrastive loss initially shows a slight increase before gradually stabilizing. Their behavior shows the distinct roles of the two objectives in jointly guiding multimodal representation learning.

## 5 CONCLUSION

In this work, we addressed the challenge of learning robust multimodal representations for medical VQA task under highly imbalanced answer distributions. We proposed HeliCon, a dual-level contrastive alignment framework that intertwines sample-level and prototype-level mechanisms with complementary hard and soft contrastive objectives. By leveraging frequent concepts as structural anchors and softly aligning rare ones, HeliCon constructs a more coherent and semantically meaningful embedding space. Experiments on multiple Med-VQA benchmarks demonstrate overall performance improvements, and ablation studies confirm the method's effectiveness in handling tail samples, highlighting the role of dual contrastive learning in enhancing robustness and generalization in medical multimodal understanding.

**Discussion.** While our experiments focus on medical VQA datasets due to the pronounced and clinically important long-tail distribution, the framework is, in principle, applicable to other long-tail multimodal problems. Because these mechanisms rely on general properties of long-tail distributions rather than domain-specific features, tail classes in other datasets can similarly benefit from relational knowledge transferred from head classes.

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

# 6 APPENDIX

## 6.1 DATASET ANSWER FREQUENCY DISTRIBUTION AND ADAPTIVE HEAD-TAIL SPLIT

Figure 7 visualizes the answer occurrence statistics for PathVQA, VQA-RAD, and SLAKE under their respective open-set training and test splits. All three datasets exhibit a long-tailed distribution, where a small number of answer categories account for the majority of samples, whereas a large portion of answers appear infrequently or even only once.

To derive a dataset-aware head-tail partition, we adopt an adaptive frequency threshold such that head classes collectively cover approximately 60% of the training samples. This avoids arbitrary cutoffs and ensures that the split naturally reflects dataset-specific distributions. The red dashed lines in Figure 7 mark the resulting thresholds. Under this criterion, PathVQA contains 142 head answers (1,950 samples) and 1,155 tail answers (1,420 samples); VQA-RAD comprises 25 head answers (70 samples) and 93 tail answers (130 samples); and SLAKE consists of 18 head answers (408 samples) and 111 tail answers (298 samples).

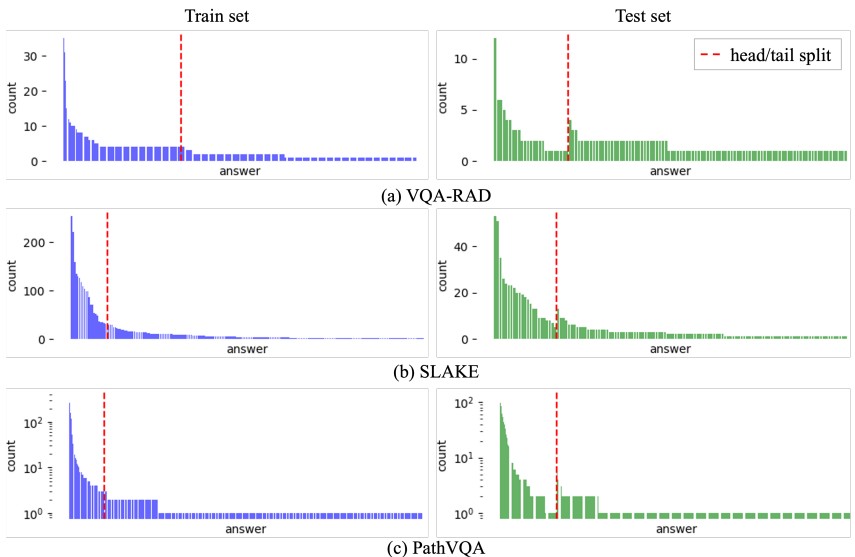

Figure 7: Answer-frequency distributions for PathVQA, VQA-RAD, and SLAKE datasets.

We further conducted experiments with different head ratios, where head samples cover 60%, 70%, or 80% of the open-set training data. Tables 5–7 present the numerical results for the three datasets. From the head/tail statistics, it can be observed that as the head ratio increases, the proportion

of unseen answer categories within the tail test set grows, making tail set prediction increasingly challenging. To better illustrate the performance comparison under different head ratios, we plotted bar charts in Figure 8. The results indicate that a head ratio of 60% generally achieves the best overall performance across three datasets. This is because increasing the head coverage introduces more fine-grained head classes with smaller sample sizes, making learning more challenging for both head and tail groups. This effect is particularly pronounced in VQA-RAD and SLAKE, where overall performance declines as the head ratio increases, while PathVQA remains relatively robust to changes in head ratio.

Table 5: Results on VQA-RAD under different head coverage ratios (60%, 70%, 80%). *H* and *T* denote head and tail sets, respectively; C/N indicates the number of answer categories (C) and the number of samples (N); *Seen* denotes test samples whose answer categories have been seen during training.

| | 60% | | | 70% | | | 80% | | |
|---|---|---|---|---|---|---|---|---|---|
| Metric | Head | Tail | Overall | Head | Tail | Overall | Head | Tail | Overall |
| Recall | 85.11 | 59.73 | 68.61 | 79.39 | 56.15 | 66.14 | 77.47 | 49.85 | 66.83 |
| F1-score | 84.70 | 61.92 | 69.89 | 78.37 | 57.72 | 66.60 | 76.24 | 51.83 | 66.85 |
| BLEU1 | 73.40 | 60.65 | 65.11 | 72.69 | 60.47 | 65.73 | 67.88 | 56.30 | 63.42 |
| Statistic | *H* C/N | *T* C/N | head ratio | *H* C/N | *T* C/N | head ratio | *H* C/N | *T* C/N | head ratio |
| Train | 138/811 | 330/540 | 60.03% | 178/946 | 290/405 | 70.02% | 246/1082 | 222/269 | 80.09% |
| Test | 25/70 | 93/130 | 35.00% | 33/86 | 85/114 | 43.00% | 52/123 | 66/77 | 61.50% |
| *Seen* | 25 | 54 | – | 33 | 46 | – | 52 | 27 | – |

Table 6: Results on SLAKE under different head coverage ratios (60%, 70%, 80%). The symbols *H*, *T*, C/N, and *Seen* are the same as described in Table 5.

| | 60% | | | 70% | | | 80% | | |
|---|---|---|---|---|---|---|---|---|---|
| Metric | Head | Tail | Overall | Head | Tail | Overall | Head | Tail | Overall |
| Recall | 97.03 | 76.91 | 88.54 | 95.52 | 72.22 | 88.13 | 91.91 | 66.46 | 86.39 |
| F1-score | 96.79 | 76.93 | 88.41 | 95.04 | 72.73 | 87.96 | 91.56 | 67.39 | 86.32 |
| BLEU1 | 90.95 | 64.04 | 79.59 | 87.94 | 62.24 | 79.78 | 82.49 | 58.69 | 77.33 |
| Statistic | *H* C/N | *T* C/N | head ratio | *H* C/N | *T* C/N | head ratio | *H* C/N | *T* C/N | head ratio |
| Train | 18/1968 | 199/1269 | 60.80% | 28/2269 | 189/968 | 70.10% | 49/2600 | 168/637 | 80.32% |
| Test | 18/408 | 111/298 | 57.79% | 28/482 | 101/224 | 68.27% | 49/553 | 80/153 | 80.29% |
| *Seen* | 18 | 109 | – | 28 | 99 | – | 49 | 78 | – |

Table 7: Results on PathVQA under different head coverage ratios (60%, 70%, 80%). The symbols *H*, *T*, C/N, and *Seen* are the same as described in Table 5.

| | 60% | | | 70% | | | 80% | | |
|---|---|---|---|---|---|---|---|---|---|
| Metric | Head | Tail | Overall | Head | Tail | Overall | Head | Tail | Overall |
| Recall | 56.47 | 23.77 | 42.69 | 54.41 | 21.85 | 42.37 | 53.02 | 20.80 | 42.17 |
| F1-score | 56.16 | 27.18 | 43.95 | 55.15 | 24.17 | 43.70 | 53.45 | 23.75 | 43.45 |
| BLEU1 | 61.07 | 55.30 | 58.64 | 57.94 | 50.55 | 55.21 | 57.41 | 51.39 | 55.38 |
| Statistic | *H* C/N | *T* C/N | head ratio | *H* C/N | *T* C/N | head ratio | *H* C/N | *T* C/N | head ratio |
| Train | 170/5973 | 3053/3976 | 60.04% | 528/6965 | 2695/2984 | 70.01% | 1234/7960 | 1989/1989 | 80.01% |
| Test | 142/1950 | 1155/1420 | 57.86% | 236/2124 | 1061/1246 | 63.03% | 295/2235 | 1002/1135 | 66.32% |
| *Seen* | 142 | 280 | – | 236 | 186 | – | 295 | 127 | – |

## 6.2 SENSITIVITY ANALYSIS ABOUT LOSS BALANCING FACTOR

Figure 9 (a) shows the training dynamics of the autoregressive, head, tail, and contrastive losses, with the head and tail losses exhibiting consistent behavior, supporting the choice of setting $\lambda_1 = \lambda_2$. Figure 9 (b) presents the results of a parameter sweep to identify the optimal value of $\lambda$. As the VQA-RAD dataset lacks a validation set, these experiments were performed only on the SLAKE and PathVQA datasets.

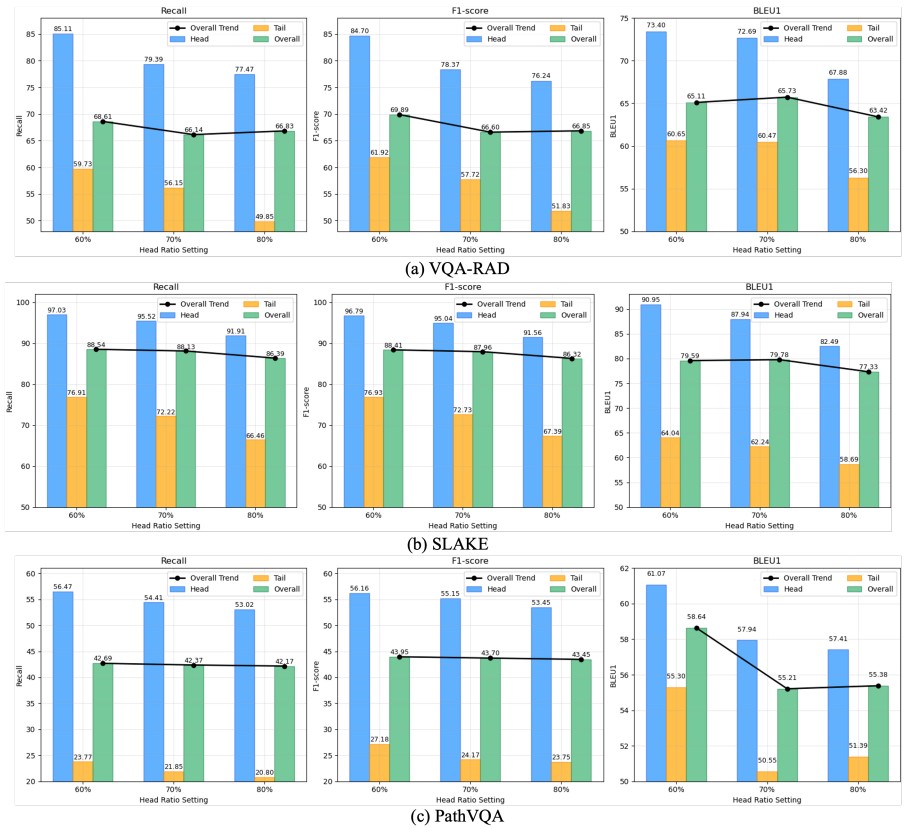

Figure 8: Performance comparison under different head-sample ratios (60%, 70%, 80%) on VQA-RAD, SLAKE, and PathVQA. Bar plots show head, tail, and overall performance across Recall, F1-score, and BLEU-1 metrics, while the line plot highlights the trend of overall performance as the head ratio increases.

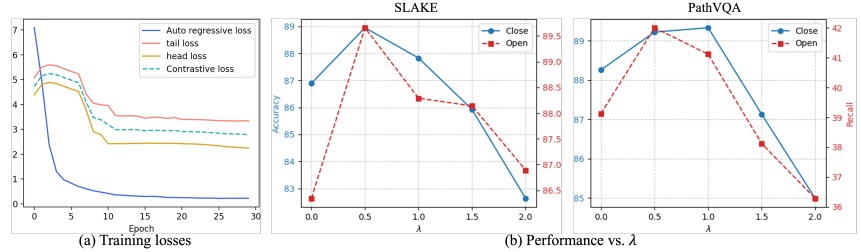

Figure 9: Training dynamics and ablation study of the balancing factors $\lambda_1$ and $\lambda_2$

## 6.3 ABLATION STUDY ON THE RAI MODULE

Figure 10 illustrates the performance improvements obtained by incorporating the RAI module across the VQA-RAD, SLAKE, and PathVQA datasets, using LLaMA-2-7B as the backbone LLM. Across all three datasets, the additional gains from RAI are relatively modest compared to the improvements brought by the dual-level contrastive learning strategy. This indicates that the majority of the performance enhancement primarily stems from the dual-level contrastive alignment mechanism.

## 6.4 THEORETICAL ANALYSIS OF USING KULLBACK–LEIBLER DIVERGENCE FOR HEAD-TAIL KNOWLEDGE TRANSFER

We first define the similarity vectors between a tail sample and head prototypes at both the input (multimodal embedding) and supervision (answer embedding) levels:

$$\mathbf{z}_j^x = \left[ x_j^\top p_1^x, \ldots, x_j^\top p_H^x \right], \quad \mathbf{z}_j^y = \left[ y_j^\top p_1^y, \ldots, y_j^\top p_H^y \right], \tag{16}$$

where $x_j \in \mathcal{B} \cap \mathcal{T}$ denotes a tail sample's multimodal input embedding, $y_j$ its corresponding answer embedding, $p_k^x$ and $p_k^y$ are the multimodal and answer prototypes of head categories $\mathcal{C}_\mathcal{H}$, with

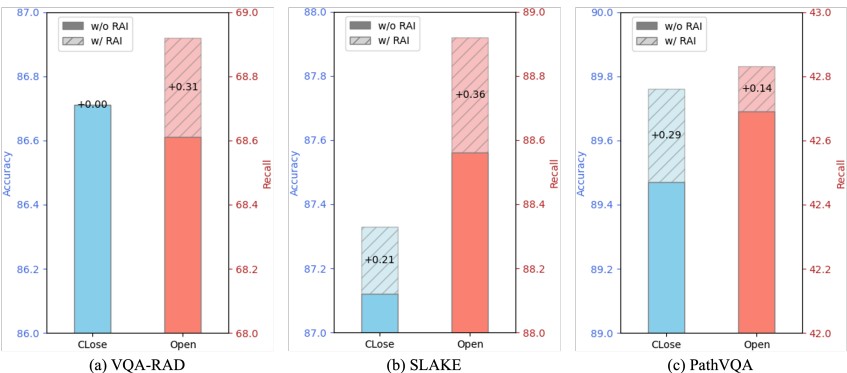

Figure 10: Ablation study of the RAI module on VQA-RAD, SLAKE, and PathVQA datasets.

$H = |\mathcal{C}_{\mathcal{H}}|$. We scale the similarities by a temperature $\tau > 0$ and normalize them using the softmax function to obtain probability distributions over the head prototypes:

$$\hat{s}_j^x = \mathrm{softmax}(\mathbf{z}_j^x / \tau), \quad s_j^y = \mathrm{softmax}(\mathbf{z}_j^y / \tau). \tag{17}$$

The prototype-level soft contrastive loss for tail embeddings is then formulated as

$$\mathcal{L}_{tail} = \sum_{x_j \in \mathcal{B} \cap \mathcal{T}} KL(s_j^y \,\|\, \hat{s}_j^x) = \sum_{x_j \in \mathcal{B} \cap \mathcal{T}} \sum_{k=1}^{H} s_{jk}^y \log \frac{s_{jk}^y}{\hat{s}_{jk}^x}. \tag{18}$$

Taking the gradient with respect to a tail embedding $x_j$, we have

$$\frac{\partial \mathcal{L}_{tail}}{\partial x_j} = \frac{1}{\tau} \sum_{k=1}^{H} (\hat{s}_{jk}^x - s_{jk}^y) p_k^x, \tag{19}$$

which explicitly shows that the tail embedding is updated as a weighted combination of head prototypes, with weights determined by the difference between predicted and target similarity distributions. If the tail multimodal embedding underestimates its similarity to a head multimodal prototype relative to the target similarity defined by the tail answer embedding and head answer prototype ($\hat{s}_{jk}^x < s_{jk}^y$), the corresponding term in the gradient is negative, moving $x_j$ closer to $p_k^x$. Conversely, if it overestimates ($\hat{s}_{jk}^x > s_{jk}^y$), the term is positive, pushing $x_j$ away from $p_k^x$.

Unlike a naive one-to-one alignment, this mechanism allows tail embeddings to inherit the relational structure among head prototypes. Specifically, the target probabilities $s_j^y$ encode the global similarity patterns of the tail sample to head answer prototypes, so aligning $\hat{s}_j^x$ to $s_j^y$ guides the tail multimodal embedding to inherit the relational structure encoded in the answer embedding space. Intuitively, the tail embedding update can be viewed as a projection onto the head embedding manifold:

$$x_j \leftarrow x_j - \eta \frac{\partial \mathcal{L}_{tail}}{\partial x_j} = x_j - \frac{\eta}{\tau} P^x \Delta_j, \tag{20}$$

where $\eta$ is the learning rate, $P^x = [p_1^x, \ldots, p_H^x] \in \mathbb{R}^{d \times H}$, and $\Delta_j = \hat{s}_j^x - s_j^y \in \mathbb{R}^H$. Over successive updates, tail embeddings converge to regions in the head embedding space that respect the head-to-head similarity structure, effectively transferring relational knowledge from head to tail embeddings.

### 6.5 COMPUTATIONAL OVERHEAD OF DUAL MEMORY BANKS

To evaluate the computational overhead introduced by dual memory banks, we measured GPU memory usage and training time per epoch across three datasets (VQA-RAD, SLAKE, and PathVQA), using Llama2-7B as the large language model (LLM). As shown in Table 8, using dual memory banks increases GPU memory consumption and training time compared to the baseline. The increase in computational overhead is modest for smaller datasets, such as VQA-RAD, but becomes more pronounced for larger datasets like PathVQA. This additional cost arises from maintaining two memory banks: one for prototype memory, which stores the prototype embeddings for both the multimodal

input and the answer, and another for sample memory, which holds the sample embeddings for the multimodal input and the answer. Despite the added computational burden, the increases remain within acceptable limits compared to the baseline model usage.

Table 8: Comparison of GPU memory usage (GB) and training time (min) with and without dual memory banks across three datasets. DMB denotes the dual memory banks, and C/N indicates the number of answer classes (C) and the number of samples (N) in the training set of each dataset.

| Setting | VQA-RAD (C/N: 470/3064) | | SLAKE (C/N: 219/4,918) | | PathVQA (C/N: 3,225/19,755) | |
|---|---|---|---|---|---|---|
| | GPU Memory | Time/Epoch | GPU Memory | Time/Epoch | GPU Memory | Time/Epoch |
| Baseline | 86.4 | 2.1 | 86.6 | 3.3 | 87.3 | 10.7 |
| +DMB | 87.1 | 2.2 | 89.3 | 3.7 | 94.8 | 11.9 |
| $\Delta$ | +0.7 | +0.1 | +2.7 | +0.4 | +7.5 | +1.2 |

