# OpenReview forum: "HeliCon: Dual-Level Contrastive Alignment for Robust Medical VQA under Long-Tailed Distribution"
_ICLR.cc/2026/Conference — Submitted to ICLR 2026_

### Official Review · Reviewer_14MG · 2025-10-28

**Soundness:** 3
**Presentation:** 3
**Contribution:** 3
**Rating:** 8
**Confidence:** 4

**Summary:**

This paper proposes HeliCon, a framework designed to improve medical visual question answering (Med-VQA) under long-tailed answer distributions, where rare clinical concepts are underrepresented and poorly learned.HeliCon introduces a dual-level contrastive alignment mechanism that enhances the robustness and semantic structure of multimodal embeddings:

Two memory banks
• an instance-level memory preserving sample diversity, and
• a prototype-level memory maintaining category-wise semantic structure.

Dual contrastive objectives
• Hard contrastive learning aligns head (frequent) concepts at both instance and prototype levels, and
• Soft contrastive learning transfers relational knowledge from head to tail (rare) concepts via KL-divergence alignment.

Retrieval-augmented inference :enriches test-time reasoning by retrieving semantically related answer embeddings from the training set and integrating them through cross-attention.

Extensive experiments on VQA-RAD, SLAKE, and PathVQA show that HeliCon consistently outperforms prior methods, achieving up to 3.51% absolute gain over the state of the art. The approach yields more structured, semantically coherent, and tail-robust embeddings, improving both answer accuracy and generalisation in medical multimodal reasoning.

**Strengths:**

Originality:
The paper introduces a dual level contrastive alignment framework that explicitly targets answer distribution bias in medical visual question answering (MedVQA), a dimension largely overlooked by prior work which has mostly focused on question bias or general VQA settings. The idea of using head (frequent) concepts as structural anchors to guide tail (rare) concept alignment through hard and soft contrastive objectives is both conceptually novel and practically valuable. The integration of instance level and prototype level memory banks provides a new, coherent perspective for balancing diversity and semantic clustering in multimodal learning.

Quality:
The methodology is technically sound and well motivated, combining principles from contrastive learning, long tailed recognition, and retrieval augmented reasoning. Each component, memory bank design, dual level contrastive losses, and retrieval based inference, is clearly defined and empirically justified. The experimental evaluation is comprehensive, spanning multiple datasets (VQA RAD, SLAKE, PathVQA), several LLM backbones (GPT2, StableLM, Mistral, LLaMA2), and detailed ablation studies that isolate the contribution of each component. The improvements are consistent, and qualitative analyses (tSNE, correlation statistics) convincingly support the claims.

Clarity:
The paper is exceptionally clear and well structured, adhering closely to ICLR standards. The motivation flows logically from the introduction to the formulation, with helpful diagrams (Figures 1 and 2) illustrating both intuition and architecture. Mathematical notation is consistent, and implementation details are sufficient for reproducibility. The inclusion of explicit training objectives, parameter settings, and model variants strengthens transparency.

Significance:
The contribution is highly relevant to both the medical AI and representation learning communities. It addresses a real world challenge, learning from imbalanced clinically meaningful data, while offering methodological insights that generalise beyond MedVQA to other multimodal or long tailed domains (for example retrieval based reasoning and grounded language models). The demonstrated 3.5 percent absolute improvement on PathVQA and consistent gains elsewhere highlight the framework’s practical utility.

Overall:
A technically rigorous, clearly written, and conceptually well integrated paper. It meaningfully extends contrastive learning to long tailed multimodal reasoning and provides both empirical and structural insights. Its combination of originality, execution quality, and broader applicability makes it a strong and significant contribution for the ICLR community.

**Weaknesses:**

1. Limited theoretical grounding of the soft alignment mechanism
The proposed prototype level soft contrastive objective, formulated as a Kullback–Leibler divergence between head and tail distributions, is empirically effective but theoretically underdeveloped. The paper does not provide a clear justification for why this formulation is the most appropriate means of transferring relational structure from head to tail concepts. A stronger theoretical analysis or connection to existing frameworks such as knowledge distillation (Hinton et al., 2015) or manifold alignment (Wang and Mahadevan, 2009) would strengthen the conceptual depth of the work. The authors could also provide an empirical comparison with alternative formulations such as Earth Mover’s Distance or cosine similarity based transfer.

2. Incomplete exploration of retrieval augmented inference
Although the retrieval component contributes to performance gains, its design and evaluation remain superficial. The retrieval mechanism is limited to a simple top K search followed by parameter free cross attention, without examining scalability or sensitivity to retrieval noise. There is no comparison with stronger retrieval based baselines such as LLaVA Med or RAG LLMs. A deeper analysis showing how retrieval quality affects answer accuracy and interpretability would make this part more convincing.

3. Lack of computational efficiency and scalability analysis
The framework maintains two memory banks updated throughout training, which could introduce nontrivial memory and time overhead, especially for large medical datasets. However, the paper does not report training time, GPU usage, or the impact of bank size on convergence. Including these results would clarify the practicality of the method in real world clinical settings.

4. Moderate novelty relative to prior contrastive works
While the integration of instance and prototype level alignment is elegant, similar multi level or hierarchical contrastive frameworks have appeared in prior work such as MUMC (Li et al., 2023b) and PMC CLIP (Lin et al., 2023). The main novelty lies in addressing answer distribution bias rather than introducing a fundamentally new learning principle. The paper could strengthen its originality by positioning HeliCon as a more general long tailed contrastive framework and empirically validating it beyond the medical VQA domain, for example on standard VQA v2 or GQA datasets.

5. Missing error and failure case analysis
While qualitative visualizations are presented, the paper does not discuss failure cases where the model retrieves misleading examples or confuses semantically similar but clinically distinct answers. Including a small number of failure analyses with attention maps or embedding plots could provide valuable insight into remaining biases and guide future improvements.

**Questions:**

Justification of the soft alignment mechanism:

Could you clarify why Kullback–Leibler divergence was chosen as the formulation for transferring head–tail relational knowledge?

Would other formulations (for example, cosine similarity or Wasserstein distance) lead to different behaviour?

A brief theoretical or empirical justification would strengthen confidence that this choice is principled rather than heuristic.

Efficiency and scalability analysis:

The dual memory banks may introduce computational overhead. Could you provide quantitative results on GPU memory usage, training time, and scalability with dataset size?

Clarifying this would help assess whether the approach is practical for large scale clinical deployment.

Retrieval augmented inference evaluation:

How is retrieval implemented and how sensitive is performance to retrieval noise or the number of retrieved samples (K)?

Could you compare your retrieval fusion strategy with standard RAG or knowledge retrieval baselines to demonstrate robustness and generality?

Generalisation beyond medical datasets:

The framework seems general for any long tailed multimodal problem. Have you considered evaluating it on a non medical dataset (for example, GQA or VQA v2) to show broader applicability?

Even a brief experiment or discussion could highlight that the method is not limited to the clinical domain.

Analysis of head–tail balance and parameter sensitivity:

The model’s performance depends on the frequency threshold separating head and tail classes and on weighting parameters λ₁ and λ₂.

Could you include or summarise a sensitivity analysis showing how performance changes with these values?

---

> ### Author Response · Authors · 2025-11-21
> **Response to Reviewer Comments**
>
> We thank the reviewer for the detailed comments and constructive suggestions. We address the concerns with detailed responses below.
>
> ---
>
> **1. Theoretical justification of the soft alignment mechanism**
>
> We chose Kullback–Leibler (KL) divergence to transfer head–tail relational knowledge, as it measures discrepancies between probability distributions over head prototypes. By aligning the predicted similarity distribution of tail embeddings with the target distribution from head embeddings, KL divergence captures global relational structure among head categories and produces weighted gradients that update tail embeddings as combinations of head prototypes. This soft alignment is crucial for tail samples, which often lack reliable positives and rely on similarity patterns learned from head prototypes.
>
> Other formulations, such as cosine similarity or the Wasserstein distance, could in principle be applied, but they are less effective at preserving the relational structure among head and tail embeddings. Empirically, we observe a clear correlation between multimodal embedding similarities and answer embedding similarities (Table 3), supporting the design choice. For a more detailed theoretical justification, we provide an analysis of how KL divergence enables effective head-to-tail knowledge transfer in Appendix 6.4 of the revised manuscript.
>
> ---
>
> **2. Analysis of training time and computational overhead**
>
> In the revised manuscript, we provide a detailed analysis of the computational overhead introduced by the dual memory banks in Appendix 6.5. Table 8 reports GPU memory usage and training time across three datasets. While the dual memory banks increase both memory consumption and training time, the overhead remains modest and acceptable relative to the baseline, demonstrating that the proposed approach is both practical and scalable to meet clinical demands.
>
> ---
>
> **3. Retrieval-augmented inference evaluation**
>
> In our method, retrieval is performed over the training data to identify answers whose sample embeddings are most similar to the input multimodal embeddings. Since we do not have an external knowledge base, RAI cannot be incorporated during training and is only applied at inference. Standard RAG or knowledge retrieval baselines typically rely on large external corpora or knowledge bases for retrieval, which are not available in our VQA setting.
>
> Regarding the number of retrieved samples, we have evaluated the impact of different numbers of retrieved samples (K). Figure 4 shows that when K = 1 or 2, the model achieves optimal performance with modest improvements. As K increases further, performance drops sharply, likely due to noisy retrieved samples negatively affecting context.
>
> ---
>
> **4. Generalization beyond medical datasets**
>
> Our experiments focus on medical VQA datasets because the long-tail distribution of answer categories is particularly pronounced and clinically important in medical domain. Due to time constraints, we have not yet conducted experiments on non-medical datasets such as GQA or VQA v2. Nevertheless, the mechanisms of HeliCon are broadly applicable, and we have included a brief discussion of this potential in the revised manuscript.
>
> ---
>
> **5. Head/tail splitting ratio and weighting parameter sensitivity analysis**
>
> We conducted experiments with different head ratios, with results for the PathVQA dataset shown below. Complete results for all datasets are provided in the revised manuscript (Tables 5–7 and Figure 8). We also evaluated different values of the weighting parameter on each dataset’s validation set, and found that 0.5 consistently yielded the best performance. This hyperparameter analysis is included in Section 4.4, with results shown in Figure 9 of Appendix 6.2.
>
> Table 1. Results under different head ratios on PathVQA.
>
> | Metric / Statistic | **60%**      |               |         | **70%**      |               |         | **80%**      |               |         |
> | ------------------ | ------------ | ------------- | ------- | ------------ | ------------- | ------- | ------------ | ------------- | ------- |
> |                    | Head         | Tail          | Overall | Head         | Tail          | Overall | Head         | Tail          | Overall |
> | Recall             | 56.47        | 23.77         | 42.69   | 54.41        | 21.85         | 42.37   | 53.02        | 20.80         | 42.17   |
> | F1-score           | 56.16        | 27.18         | 43.95   | 55.15        | 24.17         | 43.70   | 53.45        | 23.75         | 43.45   |
> | BLEU1              | 61.07        | 55.30         | 58.64   | 57.94        | 50.55         | 55.21   | 57.41        | 51.39         | 55.38   |
> | Train (H S/C/N)    | 170/5973     | 3053/3976     | 60.04%  | 528/6965     | 2695/2984     | 70.01%  | 1234/7960    | 1989/1989     | 80.01%  |
> | Test (H S/C/N)     | 142/142/1950 | 280/1155/1420 | 57.86%  | 236/236/2124 | 186/1061/1246 | 63.03%  | 295/295/2235 | 127/1002/1135 | 66.32%  |

---

### Official Review · Reviewer_zZ2j · 2025-10-29

**Soundness:** 2
**Presentation:** 3
**Contribution:** 2
**Rating:** 2
**Confidence:** 4

**Summary:**

HeliCon introduces a dual-level contrastive alignment framework for medical VQA that explicitly handles long-tailed answer distributions. It aligns frequent answers via instance- and prototype-level hard losses, while transferring relational knowledge to rare answers through soft KL alignment against head prototypes. A retrieval-augmented inference module enriches decoding with token embeddings from similar training cases. Extensive experiments show consistent gains, especially on rare diseases, and reveal a near-linear embedding space that makes rare samples retrievable and diagnostically precise.

**Strengths:**

1. The paper is well-structured and clearly written.
2. The experiments are comprehensive and the arguments are complete.

**Weaknesses:**

1. Regarding the RAI module, the paper appears to omit any dedicated ablation experiments. Moreover, in the existing ablation studies, it is never explicitly stated whether the RAI module remains active after adding SSC or PHC losses. Consequently, readers cannot clearly discern the actual contribution of the RAI module to the model’s performance.

2. The F-test in Table 3 does not provide a corresponding p-value, and significance should be indicated in bold.

**Questions:**

1. The paper does not clearly define n_thresh. My understanding is that n_thresh denotes the frequency of an answer in the dataset and is used to distinguish between head and tail samples—am I correct?

2. No ablation on the threshold itself. The three QA datasets used in the paper differ drastically in size. Fixing the threshold at 20 could plausibly cause almost all samples in PathVQA to be labeled as head and nearly all samples in VQA-RAD as tail. The authors should therefore report the exact counts of head and tail samples in each dataset. If a threshold other than 20 were used, would the multimodal embedding similarities and answer embedding similarities still fall into the same linear space?

---

> ### Author Response · Authors · 2025-11-21
> **Response to Reviewer Comments**
>
> We thank the reviewer for the careful reading of our manuscript and the constructive comments. Below we address each point in detail.
>
> ---
>
> **1. Ablation of the RAI module**
>
> We would like to clarify that the ablation experiments reported in Table 2 were conducted without the RAI module. The purpose of these experiments was to evaluate the impact of adding SSC and PHC losses independently, isolating the contribution of the dual-level contrastive learning mechanism. In the revised manuscript, we have explicitly clarified this point in the *Ablation study of contrastive learning strategies* paragraph in section 4.4.
>
> Furthermore, the contribution of the RAI module is reported in Figure 9, Appendix 6.3 in the revised manuscript. As shown, the performance gain introduced by the RAI module is relatively small compared to the improvement from the dual-level contrastive learning mechanism. This suggests that the primary performance boost comes from the proposed contrastive alignment mechanism, while RAI serves as a complementary enhancement.
>
> ---
>
> **2. F-test significance reporting**
>
> We have updated the statistical results in Table 3 to highlight significant effects in bold in the revised manuscript, where bold values indicate larger β or R² compared with the baseline.
>
> ---
>
> **3. Definition of threshold**
>
> The threshold $n_{thresh}$ denotes the answer-frequency cutoff separating head and tail samples. To account for dataset scale differences and ensure the splitting adapts to each dataset, we conducted experiments using an adaptive threshold, where head samples cover 60% of the open-set training data. Results are updated in Table 3 and added in Table 4 of the revised manuscript.
>
> ---
>
> **4. Head/tail splitting and statistics**
>
> To address the concern regarding dataset scale differences, we conducted additional experiments using an *adaptive split*, in which head samples cover 60% of the open-set training data. The updated results, including detailed head/tail statistics for all three datasets, are provided in Table 3 and Table 4 of the revised manuscript. These statistics show that tail samples contain substantially more answer categories, and that most test tail answers are unseen during training, making tail prediction inherently more challenging.
>
> Under this adaptive setting, we also re-plotted the correlation between multimodal embedding similarities and answer embedding similarities, and the figures confirm that the linear relationship is still well preserved, demonstrating that our framework is robust to the choice of threshold.
>
> We further conducted experiments using different head ratios, with head samples covering 60%, 70%, or 80% of the open-set training data. The results for two datasets are shown below, while the complete experimental results can be found in the revised manuscript (Tables 5–7) and Figure 8 in the appendix.
>
>
> Table 1. Results under different head ratios on VQA-RAD.
>
> | Metric / Statistic | **60%**  |           |         | **70%**  |           |         | **80%**   |          |         |
> | ------------------ | -------- | --------- | ------- | -------- | --------- | ------- | --------- | -------- | ------- |
> |                    | Head     | Tail      | Overall | Head     | Tail      | Overall | Head      | Tail     | Overall |
> | Recall             | 85.11    | 59.73     | 68.61   | 79.39    | 56.15     | 66.14   | 77.47     | 49.85    | 66.83   |
> | F1-score           | 84.70    | 61.92     | 69.89   | 78.37    | 57.72     | 66.60   | 76.24     | 51.83    | 66.85   |
> | BLEU1              | 73.40    | 60.65     | 65.11   | 72.69    | 60.47     | 65.73   | 67.88     | 56.30    | 63.42   |
> | Train (H C/N)    | 138/811  | 330/540   | 60.03%  | 178/946  | 290/405   | 70.02%  | 246/1082  | 222/269  | 80.09%  |
> | Test (H S/C/N)     | 25/25/70 | 54/93/130 | 35.00%  | 33/33/86 | 46/85/114 | 43.00%  | 52/52/123 | 27/66/77 | 61.50%  |
>
> Table 2. Results under different head ratios on SLAKE.
>
> | Metric / Statistic | **60%**   |             |         | **70%**   |            |         | **80%**   |           |         |
> | ------------------ | --------- | ----------- | ------- | --------- | ---------- | ------- | --------- | --------- | ------- |
> |                    | Head      | Tail        | Overall | Head      | Tail       | Overall | Head      | Tail      | Overall |
> | Recall             | 97.03     | 76.91       | 88.54   | 95.52     | 72.22      | 88.13   | 91.91     | 66.46     | 86.39   |
> | F1-score           | 96.79     | 76.93       | 88.41   | 95.04     | 72.73      | 87.96   | 91.56     | 67.39     | 86.32   |
> | BLEU1              | 90.95     | 64.04       | 79.59   | 87.94     | 62.24      | 79.78   | 82.49     | 58.69     | 77.33   |
> | Train (H S/C/N)    | 18/1968   | 199/1269    | 60.80%  | 28/2269   | 189/968    | 70.10%  | 49/2600   | 168/637   | 80.32%  |
> | Test (H S/C/N)     | 18/18/408 | 109/111/298 | 57.79%  | 28/28/482 | 99/101/224 | 68.27%  | 49/49/553 | 78/80/153 | 80.29%  |

---

### Official Review · Reviewer_mnpr · 2025-10-30

**Soundness:** 3
**Presentation:** 3
**Contribution:** 2
**Rating:** 2
**Confidence:** 5

**Summary:**

The paper tackles a key weakness in medical visual question answering (Med-VQA): models perform well on common (“head”) clinical concepts but struggle with rare (“tail”) ones that are often more medically important. This happens because existing vision–language models learn mostly from frequent examples, leading to biased or fragile representations.

The authors propose HeliCon, a new framework that makes these representations more balanced and semantically meaningful. Think of it as a “double-helix” alignment system that connects two strands: Dual memory banks, where one is for individual samples (diversity) and one for concept prototypes (structure). Dual contrastive objectives, where “hard” contrastive learning is used for frequent cases and “soft” alignment for rare ones.

Together, these mechanisms allow the model to learn effectively from both abundant and scarce examples while maintaining coherent relationships between medical concepts across the distribution.

**Strengths:**

The strength is three-fold across problem statement, experimentation, and inclusion of an ablation.
- The paper correctly identifies a real and underexplored issue in Med-VQA — long-tailed answer distributions (most prior work focused on question bias). This reframing is fresh and important.
- In experiments, HeliCon shows that this setup works well in practice. It beats strong baselines (LLaVA-Med, MUMC, Med-MoE) and shows consistent open-ended gains, particularly on PathVQA (+3.5%), which is a tough benchmark.
- The authors include ablations.

**Weaknesses:**

There are four main weakness areas here:
- HeliCon conceptually focuses on the “tails” and provides solid representational evidence that rare samples become more semantically coherent and better integrated in the embedding space. However, in practice, the main numerical gains in recall and accuracy still come from the “head” classes. The tail improvements are visible but relatively modest, suggesting that the method stabilizes learning across the distribution rather than dramatically shifting performance toward the tails.

- The paper claims improved “reasoning,” but relies only on recall and BLEU metrics for open-ended questions. There’s no reasoning-specific or factual consistency test, so the reasoning claim is overstated.

- Dual contrastive levels, memory banks, and retrieval augmentation have each been used in prior VQA or multimodal representation works (e.g., MUMC 2023b; Parashar 2024). HeliCon integrates them nicely but doesn’t introduce a fundamentally new algorithmic insight.

- Performance partly depends on the large biomedical pretraining (600 K pairs). It’s unclear how much of the gain comes from the proposed dual-contrastive method versus the pretrained initialization.

**Questions:**

You mention pretraining on 600 K image–text pairs from PMC-15M. Which components of HeliCon are included in this pretraining (e.g., visual encoder, projection layers, language model)? How much of the downstream improvement can be attributed to this initialization rather than to the proposed dual-level contrastive alignment?

The paper frequently refers to “reasoning” and “contextual reasoning,” but no reasoning-specific benchmarks or metrics are reported. Could you clarify what definition of reasoning you adopt and how the reported metrics (recall, BLEU-1, F1) reflect reasoning capability rather than lexical overlap?

Your paper emphasizes addressing the tail-side challenge in medical VQA by transferring knowledge from frequent (head) to rare (tail) concepts. However, while the embedding correlation analysis suggests stronger semantic alignment for tail samples, the quantitative gains (e.g., recall and F1 improvements) appear larger for head samples. Could you clarify whether this indicates that HeliCon primarily improves overall embedding structure rather than directly enhancing reasoning accuracy on rare concepts? In other words, how should we interpret the gap between representational improvement and task-level performance for tail samples?

---

> ### Author Response · Authors · 2025-11-21
> **Response to Reviewer Comments**
>
> We thank the reviewer for the thoughtful and detailed feedback. Our responses are as follows:
>
> ---
>
> **1. Tail improvements and task-level performance**
>
> To better illustrate HeliCon’s effectiveness on tail samples, we re-ran experiments across VQA-RAD, SLAKE, and PathVQA datasets. Updated results are presented in Tables 3 and 4 in the revised manuscript. Dataset statistics show that tail samples contain many more answer categories than head samples, and most tail answers in the test sets are **unseen during training**, whereas all head answers appear in training. This indicates that tail samples are considerably more challenging, making even modest improvements meaningful. Notably, on SLAKE, the improvement on tail samples exceeds that on head samples.
>
> Moreover, we conducted additional experiments with varying head ratios (60\%, 70\%, and 80\%). The results are presented in Tables 5–7 and illustrated in Figure 8 in the Appendix 6.1 of the revised manuscript.
>
> Regarding the TSNE visualizations in Table 2, these were generated on SLAKE to illustrate the effect of module additions on embedding clustering. Six head answer categories were selected for visualization, as head categories contain more samples and better illustrate clustering. For clarity, in the original t-SNE figures, clustering was based on predicted answer labels, and the mispredicted samples were not marked. In the updated plots, we have marked mispredicted points, demonstrating that the number of errors decreases as modules (e.g., contrastive learning strategies) are progressively added.
>
> We would also like to clarify that Table 2 (ablation study) and Table 3 (head/tail performance) are based on different datasets. To avoid any potential confusion, in the revised manuscript we have added the dataset information in the captions of each table.
>
> ---
>
> **2. Claim about reasoning**
>
> We use "reasoning" and "contextual reasoning" to refer to the model’s ability to integrate multiple pieces of information from the question and supporting text to produce a correct answer, beyond simple word matching. We acknowledge that the metrics reported in the manuscript (Recall, BLEU-1, F1-score) primarily measure overlap between predicted and reference answers, and are not reasoning-specific metrics. To avoid potential misunderstanding, statements referring to “reasoning” or “contextual reasoning” have been revised in the manuscript.
>
> ---
>
> **3. Novelty of HeliCon**
>
> While prior work such as MUMC has applied contrastive learning to medical VQA, its use is limited to pre-training in an encoder–decoder framework and relies on generic objectives like ITM/MLM. HeliCon differs fundamentally in both setting and design:
>
> 1. Feature-level contrastive learning applied on a decoder-only LLM during downstream VQA training, which is a far more challenging and rarely explored setting compared with encoder–decoder pretraining.
> 2. A dual-level, answer-conditioned contrastive objective aligns sample embeddings with both head and tail answer prototypes, enabling effective relational knowledge transfer from frequent to rare answers.
> 3. Sample- and prototype-level memory banks structure the embedding space to be semantically meaningful and robust.
>
> ---
>
> **4. Contribution beyond pretraining**
>
> To evaluate HeliCon’s contribution beyond pretrained initialization, we conducted ablation experiments comparing the baseline model, which uses pretrained parameters for initialization but is trained without dual-contrastive learning, to our full method. In Table 2, the comparison between the baseline and HeliCon on the SLAKE dataset shows that HeliCon achieves an improvement of 3.21% in recall and 1.25% in accuracy. We further summarize the improvements over the baseline across all three datasets in the table below. Results show that these gains are attributable to our proposed HeliCon method rather than merely the effect of pretrained initialization.
>
> **Table 1.** Improvements of HeliCon over the baseline on three medical VQA datasets.
>
> | Method|VQA-RAD|       |SLAKE|      |PathVQA|        |
> | -------- | ---------- | --- | ------- | ---- | --------- | ---- |
> |      |  Open | Close | Open | Close | Open | Close |
> | Baseline    | 65.80        | 84.63         | 85.71      | 86.08       | 39.57        | 88.90         |
> | Helicon     | 68.92        | 86.71         | 88.92      | 87.33       | 42.83        | 89.76         |
> | Improvement | 3.12         | 2.08          | 3.21       | 1.25        | 3.26         | 0.86          |

---

### Official Review · Reviewer_rZED · 2025-10-30

**Soundness:** 3
**Presentation:** 3
**Contribution:** 2
**Rating:** 4
**Confidence:** 4

**Summary:**

The paper introduces HeliCon, a novel dual-level contrastive learning framework for long-tailed medical visual question answering. It builds instance- and prototype-level memory banks to structure multimodal embeddings via contrastive alignment and employs a Retrieve and Fusion module during inference to recall semantically related samples, thereby improving reasoning and generalization on tail concepts.

**Strengths:**

1. They achieve high performance compared with the methods used for comparison.

2. The paper presents substantial results showing that the proposed components are effective.

3. The paper is well written, and the proposed method can be reproduced.

**Weaknesses:**

1. The paper does not include commonly used benchmark methods [A, B] for comparison, particularly those that show high zero-shot performance on the same VQA-RAD, SLAKE, and PathVQA datasets. (It seems that different metrics were used for the comparison methods, so it is difficult for the reviewer to make a fair judgment about the performance.)

[A] Chen, Junying, et al. "Towards injecting medical visual knowledge into multimodal llms at scale." Proceedings of the 2024 Conference on Empirical Methods in Natural Language Processing. 2024.

[B] Lin, Tianwei, et al. "HealthGPT: A Medical Large Vision-Language Model for Unifying Comprehension and Generation via Heterogeneous Knowledge Adaptation." Forty-second International Conference on Machine Learning.

2. The improvements in tail performance shown in Table 3 are marginal. Additionally, the performance on tail samples is particularly important for understanding the effectiveness of the proposed components. Therefore, results for all datasets on the tail samples should be provided.

3. It seems that the model is only applicable to short-answer VQA scenarios. If the answers are longer than a few words (as is common in recent datasets), can the proposed method still perform well in such cases? Otherwise, the contribution of the proposed method is too limited.

4. I acknowledge that head and tail samples are treated separately, but what is the motivation for using contrastive learning to address the long-tail problem instead of other loss functions? Further elaboration regarding this motivation is required.

5. I wonder if the model would maintain high performance in Table 2 (the ablation study on learning strategies) when RAG is excluded. I am also curious whether contrastive learning truly helps improve model learning or the similarity retrieval process in RAG.

**Questions:**

1. Provide clear information about the baselines described in the results section.

2. Does varying λ1 and λ1 impact performance, and why were they set to 0.5?

---

> ### Author Response · Authors · 2025-11-21
> **Response to Reviewer Comments**
>
> We sincerely thank the reviewer for the constructive comments. Below we provide detailed responses to each point.
>
> ---
> **1. Comparison with benchmark methods [A] and [B]**
>
> While a full comparison with [A] and [B] is not feasible due to their zero-shot generative setup, both papers report closed-ended accuracy on VQA-RAD and SLAKE. This allows us to conduct a partial comparison under the same closed-ended evaluation protocol. The results are summarized below:
>
> | Method                   | VQA-RAD | SLAKE |
> | ------------------------ | ------------------- | ----------------- |
> | Ours (Llama2-7B)         | 86.71               | 87.33             |
> | [A] HuatuoGPT-Vision-34B | 68.1                | 76.9              |
> | [B] HealthGPT-XL32 (32B) | 78.1                | 83.7              |
>
> Despite using a much smaller model (7B vs. 32B–34B), our method achieves clearly superior closed-ended accuracy. This indicates that although zero-shot medical VLMs demonstrate strong general capabilities, they do not surpass task-specific supervised training with our method.
>
> ---
>
> **2. Tail performance analysis**
>
> Table 3 compares performance on head and tail samples in the PathVQA open-set task, using a threshold of 20 to separate categories. The tail-to-head sample ratio is ≈1.08, while the tail-to-head answer category ratio is ≈26. Most tail answers (374/1,249) are unseen during training, whereas all head answers (48/48) appear in training.
>
> Although numerical improvements on tail samples are smaller, they are more meaningful due to the greater difficulty and larger number of answer categories in the tail.
>
> **Table 3.** Performance comparison between head and tail samples on PathVQA.
> *(Seen/C/N: Seen = number of answer categories present in training, C = number of answer categories, N = number of samples)*
>
> | Metric   | Baseline       |           |         | HeliCon       |           |         |
> | -------- | ------------- | --------- | ------- | ------------- | --------- | ------- |
> |          | Head          | Tail      | Overall | Head          | Tail      | Overall |
> | Recall   | 57.11         | 23.62     | 39.64   | 60.64         | 25.42     | 42.51   |
> | F1-score | 56.23         | 26.53     | 40.61   | 59.05         | 27.27     | 42.32   |
> | BLEU1    | 59.17         | 53.14     | 56.02   | 60.02         | 54.53     | 57.14   |
>
> Head (Seen/C/N: 48/48/1619), Tail (Seen/C/N: 374/1249/1751), Overall (Seen/C/N: 422/1297/3370)
>
> We also conducted experiments with an adaptive head/tail split (head covering 60% of training data) on three datasets, with updated results shown in Tables 3 and 4. Tail samples have many more categories than head samples, and most tail answers are unseen during training, while all head answers appear in training. This makes tail samples more challenging, so the observed improvements are particularly significant. Notably, on SLAKE, improvements on tail samples even exceed those on head samples.
>
> ---
>
> **3. Clarification on answer length and model generalization**
>
> Medical VQA is particularly meaningful in the medical domain, where key phrases or findings often suffice for diagnosis and downstream tasks like automated report generation. Thus, improvements on these tasks are important in clinical practice.
>
> Although SLAKE and PathVQA are mainly short-answer datasets, they include a subset of longer answers (e.g., over 10 words). Our approach does not rely on any assumption about answer length, and the relational knowledge transfer operates at the representation level and thus generalize naturally to longer. The model performs well on these longer-answer samples as well.
>
> ---
>
> **4. Motivation for contrastive learning for long-tail problem**
>
> The motivation is theoretically justified in Appendix 6.4 (Theoretical analysis of using KL divergence for head–tail knowledge transfer) in the revised manuscript.
>
> ---
>
> **5. Ablation study on RAG module**
>
> All results in Table 2 are obtained without RAG, as the ablation study isolates the effect of each learning strategy. RAG is only used in the full model (Table 1), yielding 0.36% and 0.21% improvements on SLAKE’s open and closed sets. An ablation of the RAI module across three datasets (Figure 10, Appendix 6.3) shows that the improvements mainly come from contrastive learning rather than RAG’s similarity retrieval.
>
> ---
>
> **6. Baselines description**
>
> The baseline follows the standard VQA training setup in multimodal LLMs, where the question, image features, and answer are formatted in a prompt-like concatenation. We have included this description in Section 4.4 of the revised manuscript.
>
> ---
>
> **7. Hyperparameters $\lambda_1$ and $\lambda_2$**
>
> We evaluated different loss weights on each dataset’s validation set (Figure 9, Appendix 6.2). Setting $\lambda_1 = \lambda_2 = 0.5$ consistently gave the best performance, balancing the main autoregressive objective with auxiliary losses. A detailed analysis is provided in Section 4.4 of the revised manuscript.

---

### Meta-Review · Area_Chair_z9zM · 2026-01-06

**Summary:**

The reviewers provided divergent evaluations. One reviewer (14MG) strongly supports acceptance, emphasizing the clarity of presentation, empirical results, and relevance to long-tailed multimodal learning. Another reviewer (rZED) considers the paper borderline, acknowledging solid engineering and results but raising concerns about limited tail gains, scope, and motivation. Two reviewers (mnpr and zZ2j) recommend rejection, citing concerns regarding limited tail-side improvements, overclaimed reasoning ability, insufficient ablation and analysis of key design choices (e.g., retrieval module, thresholding), and unclear separation between gains from pretraining and those from the proposed method.

After considering all reviews and the authors’ responses, I believe the paper does not yet meet the acceptance bar for this venue. Although the work is clearly written and addresses an important problem, the task-level performance gains on tail samples remain limited, and in several cases are smaller than those on head samples, weakening the central long-tail claim. In addition, the paper’s framing around improved “reasoning” is not sufficiently supported by the chosen evaluation metrics. Finally, the methodological novelty is incremental, and key design choices (e.g., soft alignment formulation, thresholding strategy, and retrieval contribution) are not analyzed in enough depth.

**Reviewer Concerns:**

The rebuttal successfully addressed several concerns, including additional experiments on tail samples, expanded ablations separating contrastive learning from retrieval, and improved explanations distinguishing gains from pretraining versus the proposed method.

However, several key concerns remain outstanding. Most notably, task-level improvements on tail samples are still modest and often smaller than head-side gains, which does not fully support the paper’s central motivation. In addition, the evaluation still lacks reasoning-specific metric, leaving the reasoning claim insufficiently validated.

**Reviewer Scores:**

Reviewer 14MG: Likely no change; would remain an accept, as the rebuttal does not alter their largely positive assessment.

Reviewer rZED: Likely a small increase (from borderline to weak accept), since several of their technical and clarity concerns were addressed.

Reviewer mnpr: Likely no change; core concerns about limited tail gains and incremental novelty remain.

Reviewer zZ2j: Likely no change; concerns regarding thresholding, retrieval contribution, and statistical analysis were only partially addressed.

---

### Decision · Program_Chairs · 2026-01-26

Reject